# Synthesis of Novel *Spiro*-Tetrahydroquinoline Derivatives and Evaluation of Their Pharmacological Effects on Wound Healing

**DOI:** 10.3390/ijms22126251

**Published:** 2021-06-10

**Authors:** Yan-Cheng Liou, Yan-An Lin, Ke Wang, Juan-Cheng Yang, Yeong-Jiunn Jang, Wenwei Lin, Yang-Chang Wu

**Affiliations:** 1Department of Chemistry, National Taiwan Normal University, Taipei 116, Taiwan; s89218456@gmail.com (Y.-C.L.); wenweilin@ntnu.edc.tw (W.L.); 2Graduate Institute of Integrated Medicine, China Medical University, Taichung 404, Taiwan; riinryxc@gmail.com (Y.-A.L.); waker603@163.com (K.W.); 3Chinese Medicine Research and Development Center, China Medical University Hospital, Taichung 404, Taiwan; qq9113054@gmail.com; 4Department of Medical Laboratory Science and Biotechnology, College of Medical and Health Science, Asia University, Taichung 41354, Taiwan

**Keywords:** tetrahydroquinoline, 1,3-indandione, *spiro*-tetrahydroquinoline, one-pot reaction, wound healing

## Abstract

A highly diastereoselective method for the synthesis of novel *spiro*-tetrahydroquinoline derivatives is reported here, using a one-pot reaction method. All compounds were characterized by ^1^H-nuclear magnetic resonance (NMR) and mass spectroscopy, and their stereo configurations were confirmed by X-ray analysis. These activities of these derivatives were then tested in human keratocyte cells. The responses of cells to treatment with selected compounds were studied using scratch analysis, and the compounds were tested in a mouse excision wound model. Three of the derivatives demonstrated significant wound-healing activities.

## 1. Introduction

Skin is the largest organ and the first defense to protect the human body [1,2]. It can be damaged by trauma, burns, skin diseases and so on. Severe skin trauma can impose physical, psychological, and economic burdens on patients. The wound healing process involves the coordination of many distinct but overlapping physiological spaces, comprising hemostasis, inflammation, epithelial cell proliferation, and tissue remodeling [3,4,5]. In elderly or diabetic patients, the risk of wound infection increased due to vascular aging and the lower tissue repair capability, which may eventually lead to chronic wounds [6]. Therefore, wound healing is one of the hot topics in skin surgery. At present, few drugs have been found with substantial abilities of promoting wound healing [7]. Actually, the quality of wound regeneration mainly depends on the efficiency of wound care [8]. In this study, we synthesized a series of novel *spiro*-tetrahydroquinoline derivatives and compared their effects in wound healing in human epidermal cells and animal models.

Quinoline derivatives have attracted both synthetic and biological chemists because of their diverse chemical and pharmacological properties, such as anticancer, antimycobacterial, anticonvulsant, anti-inflammatory and anti-cardiovascular diseases [9]. Some quinoline derivatives were found to have wound healing activity [10,11,12,13]. Tetrahydroquinoline derivatives, which are the reduced form of quinolines, were used as antibacterials, antitumor and anti-HIV agents [14]. However, there are very few reports about tetrahydroquinoline on wound healing even though their structures are so similar to each other [15]. It is a curious question of whether tetrahydroquinoline will deliver similar effects on wound healing or not.

The natural pharmacophore, *spiro*-1,3-indandione, has attracted considerable attention due to its diverse biological activities [16,17,18]. Fredericamycin A was isolated from a fermentation broth of the strain *Streptomyces griseus* and reported to demonstrate antibiotic properties [19]. Synthetic studies were also performed to obtain additional 1,3-indandione derivatives, which demonstrated antitumor and antibiotic activities [20,21]. In recent research, the *spiro*-1,3-indandione moiety has been considered to represent a valuable functional group for medicinal chemistry [22]. However, the synthesis of carbon *spiro*-1,3-indandiones remains a challenge [23,24]. Yan et al. previously synthesized *spiro*-1,3-indandiones from cyclic azomethine imine and 2-arylideneindene-1,3-dione via 1,3-dipolar cycloaddition [25]. Enders et al. used 1,4-dithiane-2,5-diol and 2-arylideneindene-1,3-dione as the starting materials to obtain *spiro*-1,3-indandione, using a squaramide-catalyzed sulfa-Michael/aldol domino reaction [26]. As a starting material that is commonly used for the generation of *spiro-*1,3-indandiones, 2-arylidene-1,3-indandiones were the focus of our recent study. In our previous research, 2-arylidene-1,3-indandiones were used to obtain enantioselective *spiro*-nitrocyclopropanes, which contained the *spiro*-1,3-indandione skeleton, catalyzed by cinchona alkoloid derivatives [27]. Indanedione-fused 2,6-disubstituted *spiro*-cyclohexanones have also been studied [28].

Recently, our team has focused on the wound-healing activities of natural products [29]. As tetrahydroquinoline and 1,3-indandione are both widely studied as pharmacophores in medicinal chemistry, we considered whether the synthesis of hybrids of tetrahydroquinoline and *spiro*-indandione residues could result in any meaningful active compounds. Song and Du have synthesized highly functionalized spirothiazolidinone tetrahydroquinolines via a squaramide-catalyzed cascade reaction [30]. Therefore, we synthesized *spiro*-tetrahydroquinolines compounds with a similar method for further study, focusing particularly on their wound-healing capabilities.

## 2. Results

The *ortho*-*N*-sulfonated aminophenyl *α,β*-unsaturated ketone **1a** and 2-benzylidene-1,3-indandione **2a** were used as model substrates for the synthesis of *spiro*-tetrahydroqunioline derivative **3a** via an *aza*-Michael/Michael reaction (Table 1). Initially, the reaction of **1a** with **2a** in the presence of DABCO furnished the desired product **3a** in 71% yield at 30 °C in *p*-xylene (Table 1, entry 1). Examination of different tertiary amines such as DMAP, Et_3_N and DIPEA as catalysts did not improve the yield of the product **3a** (Table 1, entries 2–4). It is worthy to note that all the catalysts furnished the product **3a** in excellent diastereomeric ratio (>20:1). Furthermore, various solvents like toluene, CH_2_Cl_2_, THF, EtOAc and Et_2_O were screened to optimize the reaction conditions (Table 1, entries 5–10). The results proved that aprotic solvents are the best for the reaction conditions such as toluene and CH_2_Cl_2_ (Table 1, entries 5 and 6). Considering the solubility of the starting materials **1** and **2**, we chose to test the CH_2_Cl_2_ as optimized solvent for further screening of reaction. When the reaction was carried out at 0 °C, the yield of the desired product **3a** was improved to 97%, but the reaction required longer times (38 h) compared to 30 °C (Table 1, entries 6 and 11). The reaction conditions were further evaluated by the catalyst loading and concentration of the solvent, which indicated that 5 mol% of DABCO was sufficient for the completion of the reaction in CH_2_Cl_2_ (1 mL) (Table 1, entries 12 and 13). The reaction conditions in Table 1, entry 13 were selected as the optimal conditions for further studies. The products were purified by the recrystallization, and identified by ^1^H-NMR, ^13^C-NMR and HRMS. The stereo configuration of the compound **3a** was determined by single crystal X-ray diffraction analysis as shown in Figure 1 [31].

With the optimal condition in hand, the substrate scope was further investigated. In general, all the substrates **1** and **2** with different electron-withdrawing and electron-donating R^1^ and R^2^ substituents were provided the desired *spiro*-tetrahydroquinolines in good to excellent yields (Table 2). At first, different indandione derivatives of R^2^ substituents were tested with **1a**. We noticed significant steric and electronic effects of the R^2^ substituent in the reaction outcome. Substrates with *meta*- and *para*-bromo groups as R^2^ substituents reacted well with **1a** to afford the *spiro*-tetrahydroquinoline products **3c** and **3d** in up to 83% yields within 3 h, whereas the substrate with R^2^ as *ortho*-bromo substituent was less reactive and the desired product **3b** was obtained in 71% yield in longer reaction times (12 h). It clearly indicates that the rate of the reaction is reduced by the sterically hindered substituents. We also found that the reaction rate also depends on the electronic properties of the R^2^ substituents. For example, the substrates bearing electron-withdrawing groups **2d**–**2g** subjected with **1a**, furnished the corresponding products **3d**–**3g** in high yields (81–85%) within 3 h (Table 1, entries 4–7), but the substrate bearing an electron-donating group such as 4-OMePh as R^2^ substituent **2h** resulted in the desired product **3h** in only moderate yield (68%), when prolonging the reaction time up to 29 h (Table 1, entry 8). In contrast, when substrates with heteroaryl (furyl or thienyl) groups **2i** and **2j** were employed, the reaction could not proceed to provide the products even after 24 h (Table 1, entries 9 and 10).

Furthermore, different R^1^ substituents of **1** were also tested in the reaction conditions to prepare the desired *spiro*-tetrahydroquinolines **3**. Delightfully, the substrate having aldehyde (R^1^ = H) group reacted well with **2a** to afford the corresponding product **3i** in 95% yield within 3 h (Table 1, entry 11). As similar as R^2^, substrates bearing electron-withdrawing R^1^ groups (**1c** and **1d**) more efficiently furnished the desired products **3j** and **3k** compared to the electron-donating group such as **1e** (Table 1, entries 12–14). When substrate with ester group **1f** was employed as the reactant, the corresponding product could not be found in the reaction (Table 1, entry 15). It could be understood that the second Michael addition was not efficient when the electron-rich ester is present rather than an aldehyde or ketone. In addition, 2-naphthyl group of **1g** also furnished the product **3m** in 85% yield in 6 h (Table 1, entry 16).

The results of the MTT test revealed the cell viability of human keratinocyte cells (HaCaT) treated with the 13 derived compounds as shown in Table 3. The survival rates of cells treated with **3b** were approximately 80% at all treatment concentrations (Appendix A). Cells treated with **3c** displayed a survival rate of 80% at concentrations of 6.25 and 12.5 µM, but the growth was inhibited at concentrations equal to or greater than 25 µM (Appendix A). Cells treated with **3i** displayed low survival rates at concentrations greater than 25 µM (Appendix A). Cells treated with **3l** displayed a survival rate of approximately 80% at all concentrations (Appendix A). The survival rates of cells treated with **3m** at concentrations of 6.25 and 12.5 µM were 80% and 60%, respectively. Compared with the other tested compounds, the five compounds described above had little effect on HaCaT cell growth (Appendix A), whereas the remaining compounds displayed the strong inhibition of HaCaT cell growth. The results of the remaining samples can be found in the Appendix A. Therefore, we selected the five compounds with limited effects on cell viability for use in subsequent experiments.

Based on the results of the cell viability experiments, the activities of the five selected compounds were evaluated in the scratch assay. The results shown in Figure 2 indicated that the wounds healed gradually healed within 15–18 h when the tested drugs were added to the culture medium. **3b** had a significant effect on wound healing at the concentration of 25 µM, but no significant effect was observed when the concentration was increased to 50 µM (Figure 3A). **3c** had an effect on wound healing at concentrations of 12.5 and 25 µM (Figure 3B). **3i** had a significant effect on wound healing at concentrations of 12.5 and 25 µM. Compared with the control groups, no significant effects on wound healing were observed for low concentration (6.25 µM) or high concentration (50 µM) treatment with either **3c** or **3i** (Figure 3C). **3l** and **3m** displayed no significant effects on the promotion of wound healing at any of the tested concentrations and were found to have inhibitory effects (Figure 3D,E). The optimal concentration data based on these results is shown in Figure 3F, and the compounds **3b**, **3c**, and **3i** were selected for further study in an animal model of wound closure.

Based on the results obtained from the scratch assay, wound-healing tests in mice were performed to examine the effects of wound treatment using **3b**, **3c**, and **3i** (Figure 4A). The control group was observed to shed the scab on the 9th day, and the wound repair was completed on the 13th day. In mice treated with **3b** at a concentration of 50 µM, the scab fell off on the 12th day, and the wound repair was completed on the 13th day (Figure 4B). Treatment with 6.25 µM of **3c** resulted in the shedding of the pupae on the 9th day, but the wound was not completely repaired by the 13th day (Figure 4C). When a 25 µM concentration of **3i** was applied, the pupae were shed on the 11th day, and the wound was repaired. The wound healing was completed by the 11th day, and the pupae were shed on the 7th day when the concentration of **3i** was increased to 50 µM (Figure 4D). We compared the optimal concentrations of **3b**, **3c**, and **3i** treatment based on the results of the mouse wound healing model. **3i** was demonstrated to have the best effect on wound healing at a concentration of 50 µM (Figure 4E).

## 3. Discussion

The plausible mechanism of the reaction is depicted in Scheme 1. Initially, the chalcone derivative **1** is deprotonated by tertiary amine, yielding a nitrogen-nucleophile **I** which would attack 2-arylidene-1,3-indandiones **2** to generate aza-Michael adduct **II**, through a first Michael addition (Scheme 1). The second intramolecular Michael addition upon **II** to provide **III** with both R^1^ and R^2^ in the same side and subsequent protonation and enolization would result in the spiro-tetrahydroquinolines **3** in high yields.

As shown in Scheme 1, the transition states have two isomers (*syn*- and *anti*-isomers) that can interchange with each other. In the piperidine ring, if the green hydrogen atom heads upward (*syn*-isomer), the steric effect of the sulfonamide and enone moiety was less impact, and compound **3** was more easily formed. On the contrary, green enone moiety heads upward in the *anti*-isomer which shows more steric effect and makes it more difficult to form **3′**.

In previous studies that have examined wound healing, most researchers have focused on natural products, such as extracts and extensions of Chinese herbal medicines or marine natural products. Only a few studies have examined the effects of synthetic chemicals on wound-healing outcomes. In this study, we explored the efficacy of wound healing by studying synthetic compounds and their related skeletal extensions. All compounds were preliminarily studied in human epidermal cells (HaCaT). Compound **3i** was selected because, at low concentrations, the survival rate of cells was over 80%. Compounds **3b**, **3c**, **3k**, and **3m** were shown to have less effect on the growth of HaCaT cells, based on the results of the MTT assay (Appendix A). We performed a scratch analysis and observed that compound **3i** has positive effects on wound healing. In the MTT assay, **3i** displayed the significant inhibition of cell growth at the 25 µM concentration. Due to the different outcomes observed for these two experiments, we considered whether **3i** could be effective in animal models. Compared with **3b** and **3c**, **3i** was demonstrated to represent the best compound for healing wounds on mouse skin. The wound was completely healed by the 11th day in mice treated with **3i**. Based on the results of the MTT and wound-healing assay, **3i** is thought to promote wound healing by promoting cell proliferation. In summary, we hope that the synthesis of spiro-tetrahydroqunioline derivatives might provide a new method for identifying chemicals for application in future wound-healing research.

## 4. Materials and Methods

All chemicals were analytical grade, purchased from Sigma-Aldrich (St. Louis, MO, USA), Alfa Aesar (Ward Hill, MA, USA), and Merck (Darmstadt, Germany). The purity of compounds was determined by TLC plates coated with Merck Silica gel 60 F254 (0.2 mm). Spots were observed under UV lamp or stained by dyeing agent. Infrared Spectroscopy (IR) were recorded on JASCOF/IR-5300 (Easton, MD, USA), polystyrene was used as internal standard to mark 1601 cm^−1^. ^1^H-NMR and ^13^C-NMR spectra were recorded on Bruker Avance 400 MHz spectrophotometer (Billerica, MA, USA). In ^1^H-NMR, CDCl_3_ was used as d-solvent, TMS as internal standard to mark 0 ppm. The definition of splitting term: singlet (s), doublet (d), triplet (t), qutratet (q), multiplet (m), coupling constant (*J*). In ^13^C-NMR, chloroform was used as internal standard to mark 77.0 ppm. Mass Spectroscopy (MS) and High-Resolution Mass Spectroscopy (HRMS) were recorded on JMS-700 (JEOL), (Tokyo, Japan), double focusing mass spectrometer (FAB and EI), Applied Biosystems 4800 Proteomics Analyze (MALDI) (Foster City, CA, USA) or Waters (Milford, MA, USA) LCT Premier XE (ESI). X-ray spectra were recorded on Enraf-Nonius FR590 and Nonious CAD4 Kappa Axis XRD (Bruker). Dulbecco’s Modified Eagle Medium (DMEM) and Penicillin-Streptomycin were purchased from Gibco (Waltham, MA, USA). Fetal Bovine Serum (FBS) was purchased from HyClone (Marlborough, MA, USA). Methythiazolyltetrazolium (MTT), Polyethylene glycol 400 (PEG 400), and Polyethylene glycol 4000 (PEG 4000) were purchased from Sigma-Aldrich. Culture insert for migration assay was purchased from ibidi (Planegg, Germany). Anesthesia for animal test (Zoletil) was purchased from Virbac (Carros, France). Fluorescence interpretation and image analysis were collected using Cytation^TM^ 5 Cell Imaging Multi-Mode Reader, BioTek (Winooski, VT, USA).

### 4.1. Ortho-N-Protected Aminophenyl α,β-Unsaturated Ketones (**1**)

2-aminobenzyl alcohol (20 mmol, 2.46 g) was dissolved in 100 mL DCM, and benzenesulfonyl chloride (22 mmol, 4.18 g) and 1 mL pyridine were added. The mixture was stirred for 12 h at r.t. The solvent was removed under reduced pressure. The residue **4** was not purified and dissolved in 50 mL DCM. Pyridinium chlorochromate (30 mmol, 6.46 g) was added and the resulting solution was stirred at r.t. for 4 h. The reaction mixture was filtered by Celite 545 and washed by DCM. After solvent was removed under reduced pressure, the residue was purified by flash chromatography (DCM: Hexanes = 2:1) to give **5**, yield 97%. **5** (3 mmol, 0.83 g) was dissolved in 15 mL toluene, **6** (3.3 mmol) was added. The mixture was stirred at 80 °C for 12 h. The solvent was removed under reduced pressure, residue was purified by flash chromatography to yield **1** as shown in Scheme 2.

(*E*)-N-(2-(3-oxo-3-phenylprop-1-en-1-yl)phenyl)benzenesulfonamide (**1a**) 

(mp 147.3–148.2 °C):

**^1^H-NMR (400 MHz, CDCl_3_, 25 °C) δ/ppm:** 7.93 (d, 2H, J = 8.2 Hz), 7.74 (d, 1H, J = 15.8 Hz), 7.71–7.66 (m, 2H),7.65–7.55 (m, 3H), 7.52–7.45 (m, 3H), 7.39 (t, 1H, J = 7.8 Hz), 7.36–7.21 (m, 4H), 7.20 (d, 1H, J = 15.8 Hz).

**^13^C-NMR (100 MHz, CDCl_3_, 25 °C) δ/ppm:** 190.21, 139.1, 138.8, 137.6, 135.2, 133.1, 132.9, 131.1, 131.0, 129.0, 128.63, 128.60, 127.9, 127.3, 127.2, 127.1, 124.2.

HRMS (ESI) for C_21_H_18_NO_3_S, [M + H]^+^ (364.1002), found 364.1006.

IR (KBr) ῡ (cm^−1^): 3236, 3063, 2821, 1655, 1602, 1573, 1333, 1218, 1165, 1091, 734.

(*E*)-N-(2-(3-oxoprop-1-en-1-yl)phenyl)benzenesulfonamide (**1b**).

mp: 137.0–137.6 °C.

**^1^H-NMR (400 MHz, CDCl_3_, 25 °C) δ/ppm:** 9.50 (d, 1H, J = 7.6 Hz), 7.71 (d, 2H, J = 7.8 Hz), 7.64–7.54 (m, 3H), 7.46 (t, 2H, J = 7.8 Hz), 7.38–7.30 (m, 2H), 7.16–7.07 (m, 2H), 6.54 (dd, 1H, J^1^ = 15.8 Hz, J^2^ = 7.8 Hz).

**^13^C-NMR (100 MHz, CDCl_3_, 25 °C) δ/ppm:** 193.8, 147.2, 138.5, 134.5, 133.4, 131.7, 131.4, 129.9, 129.2, 128.3, 128.0, 127.4, 127.3.

**IR** (KBr) ῡ (cm^−1^): 3237, 3063, 2828, 1671, 1623, 1448, 1331, 1160, 1132, 1091, 972, 758.

**HRMS** (ESI) for C_15_H_13_NO_3_SNa, [M + Na] + (310.0508), found 310.0517. 

(*E*)-N-(2-(3-(4-bromophenyl)-3-oxoprop-1-en-1-yl)phenyl)benzenesulfonamide (**1c**).

mp: 137.0–137.6 °C.

**^1^H-NMR (400 MHz, CDCl_3_, 25 °C) δ/ppm:** 7.82–7.72 (m, 3H), 7.70 (d, 2H, J = 7.8 Hz), 7.63–7.58 (m, 3H), 7.55 (s, 1H), 7.45–7.26 (m, 6H), 7.16 (d, 1H, J = 15.8 Hz).

**^13^C-NMR (100 MHz, CDCl_3_, 25 °C) δ/ppm:** 189.2, 139.8, 138.9, 136.3, 135.2, 133.0, 131.9, 131.3, 131.0, 130.1, 129.0, 128.2, 127.8, 127.3, 127.2, 123.7.

**IR** (KBr) ῡ (cm^−1^): 3236, 3072, 2835, 1654, 1598, 1397, 1330, 1216, 1164, 1091, 758.

**HRMS** (ESI) for C_21_H_17_NO_3_S^79^Br, [M + H]^+^ (442.0107), found 442.0111.

**HRMS** (ESI) for C_21_H_17_NO_3_S^81^Br, [M + H]^+^ (444.0087), found 442.0093.

(*E*)-N-(2-(3-(4-nitrophenyl)-3-oxoprop-1-en-1-yl)phenyl)benzenesulfonamide (**1d**).

mp: 185.4–185.7 °C.

**^1^H-NMR (400 MHz, CDCl_3_, 25 °C) δ/ppm:** 8.37 (d, 2H, J = 8.8 Hz), 8.13 (d, 2H, J = 8.8 Hz), 7.81 (d, 1H, J = 15.8 Hz), 7.75–7.68 (m, 3H), 7.54–7.34 (m, 6H), 7.27–7.24 (m, 1H), 6.73 (s, 1H).

**^13^C-NMR (100 MHz, CDCl_3_, 25 °C) δ/ppm:** 189.0, 150.2, 142.5, 141.1, 138.7, 135.1, 133.2, 131.7, 131.0, 129.6, 129.2, 127.7, 127.4, 127.3, 123.9.

**IR** (KBr) ῡ (cm^−1^): 3253, 3068, 2926, 1664, 1597, 1523, 1329, 1212, 1162, 1090, 734.

**HRMS** (ESI) for C_21_H_17_N_2_O_5_S, [M + H]^+^ (409.0853), found 409.0856.

(*E*)-N-(2-(3-(4-methoxyphenyl)-3-oxoprop-1-en-1-yl)phenyl)benzenesulfonamide (**1e**).

mp: 171.8–172.0 °C.

**^1^H-NMR (400 MHz, CDCl_3_, 25 °C) δ/ppm:** 7.95 (d, 2H, J = 7.8 Hz), 7.83 (s, 1H), 7.75 (d, 1H, J = 15.8 Hz), 7.71–7.65 (m, 2H), 7.59–7.50 (m, 2H), 7.39 (t, 1H, J = 7.8 Hz), 7.35–7.23 (m, 4H), 7.18 (d, 1H, J = 15.8 Hz), 6.95 (d, 2H, J = 7.8 Hz), 3.88 (s, 3H).

**^13^C-NMR (100 MHz, CDCl_3_, 25 °C) δ/ppm:** 188.4, 163.6, 138.9, 138.3, 135.1, 132.9, 131.1, 131.0, 130.9, 130.5, 129.0, 127.8, 127.2, 127.1, 124.2, 113.9, 55.5.

**IR** (KBr) ῡ (cm^−1^): 3182, 3068, 2840, 1651, 1603, 1335, 1263, 1223, 1167, 1091, 1022, 760.

**HRMS** (ESI) for C_22_H_20_NO_4_S, [M + H]^+^ (394.1108), found 394.1114.

Ethyl-(*E*)-3-(2-(phenylsulfonamido)phenyl)acrylate (**1f**).

mp: 125.1–125.6 °C.

**^1^H-NMR (400 MHz, CDCl_3_, 25 °C) δ/ppm:** 7.68 (d, 2H, J = 8.0 Hz), 7.56 (d, 1H, J = 15.8 Hz), 7.50 (t, 1H, J = 7.6 Hz), 7.47–7.42 (m, 2H), 7.42–7.33 (m, 3H), 7.27–7.21 (m, 2H), 6.12 (d, 1H, J = 15.8 Hz), 4.23 (q, 2H, J = 7.2 Hz), 1.32 (t, 3H, J = 7.2 Hz).

**^13^C-NMR (100 MHz, CDCl_3_, 25 °C) δ/ppm:** 166.5, 138.8, 134.5, 133.0, 130.9, 130.4, 129.0, 127.5, 127.3, 127.2, 127.0, 120.7, 60.8, 14.2.

**IR** (KBr) ῡ (cm^−1^): 3244, 3068, 2983, 1692, 1635, 1448, 1320, 1166, 1091, 979, 762. **HRMS** (ESI) for C_17_H_17_NO_4_SNa, [M + Na]+ (354.0770), found 354.0775.

(*E*)-N-(2-(3-(naphthalen-2-yl)-3-oxoprop-1-en-1-yl)phenyl)benzenesulfonamide (**1g**).

mp: 161.2–161.5 °C.

**^1^H-NMR (400 MHz, CDCl_3_, 25 °C) δ/ppm:** 8.45 (s, 1H), 8.02–7.94 (m, 2H), 7.91–7.84 (m, 2H), 7.77 (d, 1H, J = 15.8 Hz), 7.73–7.67 (m, 2H), 7.67–7.53 (m, 3H), 7.51–7.43 (m, 2H), 7.43–7.25 (m, 6H).

**^13^C-NMR (100 MHz, CDCl_3_, 25 °C) δ/ppm:** 190.0, 139.0, 138.9, 135.6, 135.2, 134.9, 133.0, 132.5, 131.2, 131.1, 130.4, 129.6, 129.1, 128.7, 128.6, 127.8, 127.3, 127.2, 126.9, 124.5, 124.4.

**IR** (KBr) ῡ (cm^−1^): 3213, 3063, 2813, 1653, 1626, 1598, 1485, 1327, 1165, 1091, 759.

**HRMS** (ESI) for C_25_H_20_NO_3_S, [M + H]^+^ (414.1158), found 414.1164.

### 4.2. 2-Arylidene-1,3-Indandiones (**2**)

A solution of the mixture of 1,3-indandione (5 mmol, 0.73 g), aldehyde (5.5 mmol) and L-proline (1.5 mmol, 0.17 g) in methanol (90 mL) was stirred for 12 h at r.t. The solvent was removed under reduced pressure, and the residue was washed by methanol. The solid **2** was purified by flash chromatography as shown in Scheme 3.

### 4.3. Spiro-Tetrahydroquinoline (**3**)

Compounds **1** (0.2 mmol) and **2** (0.2 mmol) were dissolved in 1 mL DCM, 1.2 mg DABCO (5 mol%) was added as catalysis. The reaction was sealed and stirred at 30 °C. After the reaction completed, the mixture was quenched with 1 N hydrochloric acid aqueous solution and extracted with DCM. The organic layer was washed with water, dried over MgSO_4_, filtered, re-crystallized in ethanol and hexane.

**3a** (110.0 mg, yield 92%, mp 243–244 °C): **1a** (72.7 mg, 0.2 mmol) and **2a** (46.9 mg, 0.2 mmol) as starting material, reacted for 12 h.

**^1^H-NMR (400 MHz, CDCl_3_, 25 °C) δ/ppm:** 8.03 (d, 1H, J = 7.8 Hz), 7.92 (d, 1H, J = 7.6 Hz), 7.78–7.69 (m, 3H), 7.69–7.60 (m, 3H), 7.59–7.44 (m, 6H), 7.37 (t, 2H, J = 7.8 Hz), 7.20 (t, 1H, J = 7.6 Hz), 7.07–6.94 (m, 5H), 6.68 (d, 1H, J = 8.0 Hz), 5.87 (s, 1H), 3.05 (d, 1H, J = 11 Hz), 2.95 (dd, 1H, J^1^ = 16.4 Hz, J^2^ = 11 Hz), 2.34 (dd, 1H, J^1^ = 16.4 Hz, J^2^ = 2.0 Hz).

**^13^C-NMR (100 MHz, CDCl_3_, 25 °C) δ/ppm:** 201.2, 197.2, 195.1, 142.7, 141.3, 138.7, 137.6, 137.3, 136.5, 135.9, 135.8, 133.5, 133.4, 133.3, 129.2, 128.7, 128.1, 127.9, 127.8, 127.7, 127.5, 127.1, 126.8, 125.3, 123.03, 122.98, 66.1, 66.0, 37.3, 36.3.

**HRMS** (ESI) for C_37_H_27_NO_5_S, [M + Na]^+^ (620.1508), found 620.1509.

**IR** (KBr) ῡ (cm^−1^): 3440, 1741, 1705, 1448, 1356, 1250, 1169, 1093, 753, 582.

**3b** (96.1 mg, yield 71%, mp 229–230 °C): **1a** (72.7 mg, 0.2 mmol) and **2b** (62.3 mg, 0.2 mmol) as starting material, reacted for 24 h.

**^1^H-NMR (400 MHz, CDCl_3_, 25 °C) δ/ppm:** 7.97 (d, 1H, J = 7.6 Hz), 7.92 (dd, 1H, J^1^ = 7.6 Hz, J^2^ = 1.2Hz), 7.84–7.79 (m, 2H), 7.76(td, 1H, J^1^ = 7.2 Hz, J^2^ = 0.8 Hz), 7.71–7.67 (m, 2H), 7.64 (td, 1H, J^1^ = 7.6 Hz, J^2^ = 0.8 Hz), 7.62–7.55 (m, 3H), 7.52 (t, 1H, J = 7.6 Hz), 7.47 (t, 1H, J = 7.6 Hz), 7.43–7.34 (m, 3H), 7.28–7.20 (m, 2H), 7.14–7.06 (m, 2H), 6.89–6.82 (m, 1H), 6.77 (d, 1H, J = 7.8 Hz), 6.29 (s, 1H), 3.14–3.00 (m, 2H), 2.33 (d, 1H, J = 16 Hz).

**^13^C-NMR (100 MHz, CDCl_3_, 25 °C) δ/ppm:** 199.9, 197.7, 195.2, 142.4, 141.7, 138.6, 137.9, 137.0, 136.3, 135.9, 134.1, 133.4, 132.0, 131.5, 129.4, 129.0, 128.6, 128.04, 127.92, 127.7, 127.4, 127.2, 127.0, 125.6, 123.4, 122.7, 121.7, 66.1, 63.3, 36.3, 36.0.

**HRMS** (ESI) for C_37_H_26_Br^79^NO_5_S, [M + Na]^+^ (698.0607), found 698.0612.

**HRMS** (ESI) for C_37_H_26_Br^81^NO_5_S, [M + Na]^+^ (700.0587), found 700.0599.

**IR** (KBr) ῡ (cm^−1^): 3066, 1742, 1707, 1594, 1448, 1358, 1251, 1171, 1090, 949, 754.

**3c** (109.6 mg, yield 81%, mp 254–255 °C): **1a** (72.7 mg, 0.2 mmol) and **2c** (62.3 mg, 0.2 mmol) as starting material, reacted for 3 h.

**^1^H-NMR (400 MHz, CDCl_3_, 25 °C) δ/ppm:** 8.03 (d, 1H, J = 7.8 Hz), 7.95 (d, 1H, J = 7.8 Hz), 7.79 (t, 1H, J = 7.4 Hz), 7.76–7.67 (m, 3H), 7.65–7.46 (m, 8H), 7.37 (t, 2H, J = 7.4 Hz), 7.21 (t, 1H, J = 7.4 Hz), 7.17–7.08 (m, 2H), 7.04 (d, 1H, J = 7.4 Hz), 6.92 (t, 1H, J = 7.6 Hz), 6.66 (d, 1H, J = 7.6 Hz), 5.80 (s, 1H), 3.08–2.87 (m, 2H), 2.33 (d, 1H, J = 16.0 Hz).

**^13^C-NMR (100 MHz, CDCl_3_, 25 °C) δ/ppm:** 200.9, 196.8, 194.9, 142.6, 141.2, 141.1, 137.3, 137.1, 136.7, 136.1, 135.8, 133.5, 133.3, 130.8, 129.80, 129.77, 129.30, 128.7, 128.0, 127.9, 127.7, 127.1, 126.9, 125.6, 125.3, 123.2, 123.1, 122.1, 65.8, 65.3, 37.4, 36.3.

**HRMS** (ESI) for C_37_H_26_Br^79^NO_5_S, [M + Na]^+^ (698.0607), found 698.0613.

**HRMS** (ESI) for C_37_H_26_Br^81^NO_5_S, [M + Na]^+^ (700.0587), found 700.0601.

**IR** (KBr) ῡ (cm^−1^): 3064, 1741, 1707, 1594, 1448, 1357, 1250, 1169, 1090, 915, 732.

**3d** (112.3 mg, yield 83%, mp 249–250 °C): **1a** (72.7 mg, 0.2 mmol) and **2d** (62.3 mg, 0.2 mmol) as starting material, reacted for 3 h.

**^1^H-NMR (400 MHz, CDCl_3_, 25 °C) δ/ppm:** 8.01 (d, 1H, J = 8.0 Hz), 7.92 (d, 1H, J = 7.6 Hz), 7.80 (t, 1H, J = 7.6 Hz), 7.75–7.66 (m, 3H), 7.64–7.44 (m, 8H), 7.36 (t, 2H, J = 7.6 Hz), 7.20 (t, 1H, J = 7.6 Hz), 7.15 (d, 2H, J = 8.4 Hz), 6.95 (d, 2H, J = 8.4 Hz), 6.65 (d, 1H, J = 7.4 Hz), 5.83 (s, 1H), 3.02 (d, 1H, J = 11 Hz), 2.92 (dd, 1H, J^1^ = 16.4 Hz, J^2^ = 11 Hz), 2.32 (dd, 1H, J^1^ = 16.4 Hz, J^2^ = 1.8 Hz). 

**^13^C-NMR (100 MHz, CDCl_3_, 25 °C) δ/ppm:** 200.8, 197.0, 194.9, 142.5, 141.2, 138.0, 137.4, 137.2, 136.8, 136.2, 135.8, 133.45, 133.40, 133.2, 131.3, 129.3, 128.7, 128.6, 127.9, 127.8, 127.6, 127.1, 126.9125.3, 123.3, 123.1, 121.6, 65.7, 65.3, 37.6, 36.2.

**HRMS** (ESI) for C_37_H_26_Br^79^NO_5_S, [M + Na]^+^ (698.0607), found 698.0610.

**HRMS** (ESI) for C_37_H_26_Br^81^NO_5_S, [M + Na]^+^ (700.0587), found 700.0598.

**IR** (KBr) ῡ (cm^−1^): 3068, 1741, 1707, 1594, 1488, 1356, 1250, 1169, 1073, 753.

**3e** (108.7 mg, yield 86%, mp 253–254 °C): **1a** (72.7 mg, 0.2 mmol) and **2e** (53.7 mg, 0.2 mmol) as starting material, reacted for 3 h.

**^1^H-NMR (400 MHz, CDCl_3_, 25 °C) δ/ppm:** 8.01 (d, 1H, J = 7.8 Hz), 7.92 (d, 1H, J = 7.8 Hz), 7.78 (td, 1H, J^1^ = 7.6 Hz, J^2^ = 0.8 Hz), 7.75–7.67 (m, 3H), 7.65–7.44 (m, 8H), 7.36 (t, 2H, J = 7.6 Hz), 7.20 (t, 1H, J = 7.6 Hz), 7.04–6.96 (m, 4H), 6.66 (d, 1H, J = 7.6 Hz), 5.84 (s, 1H), 3.02 (d, 1H, J = 11 Hz), 2.93 (dd, 1H, J^1^ = 16.4 Hz, J^2^ = 11 Hz), 2.32 (dd, 1H, J^1^ = 16.4 Hz, J^2^ = 1.6 Hz).

**^13^C-NMR (100 MHz, CDCl_3_, 25 °C) δ/ppm:** 200.9, 197.0, 194.9, 142.6, 141.2, 137.5, 137.4, 137.2, 136.7, 136.1, 135.8, 133.5, 133.4, 133.2, 129.3, 128.7, 128.4, 128.3, 127.9, 127.8, 127.6127.1, 126.9, 125.3, 123.3, 123.1, 65.8, 65.3, 37.5, 36.2.

**HRMS** (ESI) for C_37_H_26_Cl^35^NO_5_S, [M + Na]^+^ (654.1112), found 654.1118.

**HRMS** (ESI) for C_37_H_26_Cl^37^NO_5_S, [M + Na]^+^ (655.1146), found 656.1104.

**IR** (KBr) ῡ (cm^−1^): 3112, 1739, 1707, 1583, 1355, 1250, 1170, 752.

**3f** (100.9 mg, yield 81%, mp 239–240 °C): **1a** (72.7 mg, 0.2 mmol) and **2f** (51.9 mg, 0.2 mmol) as starting material, reacted for 3 h.

**^1^H-NMR (400 MHz, CDCl_3_, 25 °C) δ/ppm:** 8.04 (d, 1H, J = 8.0 Hz), 7.94 (d, 1H, J = 7.8 Hz), 7.81 (t, 1H, J = 7.4Hz), 7.74 (t, 1H, J = 7.6 Hz), 7.70 (d, 2H, J = 7.6 Hz), 7.65–7.58 (m, 3H), 7.58–7.46 (m, 5H), 7.42–7.30 (m, 4H), 7.24–7.16 (m, 3H), 6.65 (d, 1H, J = 7.8 Hz), 5.91 (s, 1H), 3.03 (d, 1H, J = 11 Hz), 2.93 (dd, 1H, J^1^ = 16.4 Hz, J^2^ = 11 Hz), 2.32 (dd, 1H, J^1^ = 16.4 Hz, J^2^ = 1.6 Hz).

**^13^C-NMR (100 MHz, CDCl_3_, 25 °C) δ/ppm:** 200.4, 196.7, 194.7, 144.3, 142.3, 141.1, 137.1, 136.99, 136.96, 136.4, 135.7, 133.6, 133.5, 132.9, 132.0, 129.4, 128.7, 128.1, 127.8, 127.6, 127.5, 127.1, 125.3, 123.29, 123.23, 118.3, 111.5, 65.6, 65.2,37.7, 36.1.

**HRMS** (ESI) for C_38_H_26_N_2_O_5_S, [M + Na]^+^ (645.1455), found 645.1459.

**IR** (KBr) ῡ (cm^−1^): 3485, 2356, 1736, 1707, 1360, 1250, 1170, 749, 581.

**3g** (105.4 mg, yield 82%, mp 247–248 °C): **1a** (72.7 mg, 0.2 mmol) and **2g** (55.9 mg, 0.2 mmol) as starting material, reacted for 3 h.

**^1^H-NMR (400 MHz, CDCl_3_, 25 °C) δ/ppm:** 8.06 (d, 1H, J = 8.0 Hz), 7.95 (d, 1H, J = 7.6 Hz), 7.91 (d, 2H, J = 8.6Hz), 7.81 (t, 1H, J = 7.4 Hz), 7.76–7.67 (m, 3H), 7.65–7.58 (m, 3H), 7.58–7.47 (m, 5H), 7.36 (t, 2H, J = 7.8 Hz), 7.31–7.25 (m, 2H), 7.22 (t, 1H, J = 7.6 Hz), 6.66 (d, 1H, J = 7.6 Hz), 5.97 (s, 1H), 3.04 (d, 1H, J = 11 Hz), 2.94 (dd, 1H, J^1^ = 16.4 Hz, J^2^ = 11 Hz), 2.33 (dd, 1H, J^1^ = 16.4 Hz, J^2^ = 1.8 Hz).

**^13^C-NMR (100 MHz, CDCl_3_, 25 °C) δ/ppm:** 200.3196.5, 194.6, 1478.1, 146.3, 142.2, 141.0, 137.1, 137.0, 136.9, 136.5, 135.7, 133.6, 133.5, 132.8, 129.4, 128.7, 128.1, 127.9, 127.8, 127.5, 127.1, 125.3, 123.4, 123.31, 123.26, 65.6, 64.9, 37.8, 36.1.

**HRMS** (ESI) for C_37_H_26_N_2_O_7_S, [M + Na] ^+^ (665.1353), found 665.1359.

**IR** (KBr) ῡ (cm^−1^): 3068, 1743, 1708, 1598, 1522, 1340, 1250, 1169, 1088, 689.

**3h** (85.4 mg, yield 68%, mp 241–242 °C): **1a** (72.7 mg, 0.2 mmol) and **2h** (52.9 mg, 0.2 mmol) as starting material, reacted for 29 h.

**^1^H-NMR (400 MHz, CDCl_3_, 25 °C) δ/ppm:** 8.00 (dd, 1H, J^1^ = 8.0 Hz, J^2^ = 1.2 Hz), 7.91 (d, 1H, J = 7.6 Hz), 7.75 (td, 1H, J^1^ = 7.2 Hz, J^2^ = 1.2 Hz), 7.72–7.60 (m, 5H), 7.59–7.43 (m, 6H), 7.36 (t, 2H, J = 7.6 Hz), 7.19 (td, 1H, J^1^ = 7.8 Hz, J^2^ = 1.2 Hz), 6.96 (d, 2H, J = 8.6 Hz), 6.67 (d, 1H, J = 7.6 Hz), 6.54 (d, 2H, J = 8.6 Hz), 5.83 (s, 1H), 3.61 (s, 3H), 3.04 (d, 1H, J = 11 Hz), 2.93 (dd, 1H, J^1^ = 16.4 Hz, J^2^ = 11 Hz), 2.33 (dd, 1H, J^1^ = 16.4 Hz, J^2^ = 2.0 Hz).

**^13^C-NMR (100 MHz, CDCl_3_, 25 °C) δ/ppm:** 201.3, 197.3, 195.1, 158.7, 142.8, 141.3, 137.6, 137.4, 136.5, 135.85, 135.8 3, 133.6, 133.4, 133.2, 131.0, 129.2, 128.6, 128.0, 127.9, 127.74, 127.68, 127.1, 126.7, 125.3, 123.1, 123.0, 113.5, 66.1, 65.7, 55.0, 37.3, 36.3.

**HRMS** (ESI) for C_38_H_29_NO_6_S, [M + Na]^+^ (650.1608), found 650.1611.

**IR** (KBr) ῡ (cm^−1^): 3068, 1739, 1706, 1511, 1353, 1249, 1172, 1086, 720.

**3i** (99.1 mg, yield 95%, mp 204–205 °C): **1c** (54.5 mg, 0.2 mmol) and **2a** (46.9 mg, 0.2 mmol) as starting material, reacted for 3 h.

**^1^H-NMR (400 MHz, CDCl_3_, 25 °C) δ/ppm:** 9.13 (d, 1H, J = 2.4 Hz), 8.04 (dd, 1H, J^1^ = 8.0 Hz, J^2^ = 0.8 Hz), 7.92 (d, 1H, J = 7.6 Hz), 7.75 (td, 1H, J^1^ = 7.6 Hz, J^2^ = 0.8 Hz), 7.73–7.68 (m, 2H), 7.67–7.61 (m, 2H), 7.57–7.51 (m, 3H), 7.41 (d, 1H, J = 7.6 Hz), 7.30 (td, 1H, J^1^ = 7.6 Hz, J^2^ = 0.8 Hz), 7.04–6.96 (m, 5H), 6.68 (d, 1H, J = 7.8 Hz), 5.91 (s, 1H), 2.61 (dd, 1H, J^1^ = 11.8 Hz, J^2^ = 2.4 Hz), 2.25 (ddd, 1H, J^1^ = 17.4 Hz, J^2^ = 11.8 Hz, J^3^ = 2.4 Hz), 1.83(dd, 1H, J^1^ = 17.4 Hz, J^2^ = 2.4 Hz).

**^13^C-NMR (100 MHz, CDCl_3_, 25 °C) δ/ppm:** 200.9, 198.1, 196.5, 124.6, 141.1, 138.4, 137.7, 137.4, 136.6, 136.0, 133.2, 133.0, 129.1, 128.4, 128.3, 128.2, 127.6, 127.3, 127.1, 126.7, 125.0, 123.1, 123.0, 65.9, 65.8, 40.9, 36.2.

**HRMS** (ESI) for C_31_H_23_NO_5_S, [M + Na]^+^ (544.1189), found 544.1194.

**IR** (KBr) ῡ (cm^−1^): 3059, 2861, 1722, 1710, 1585, 1357, 1246, 1169, 1083, 704.

**3j** (125.9 mg, yield 93%, mp 265–266 °C): **1c** (88.5 mg, 0.2 mmol) and **2a** (46.9 mg, 0.2 mmol) as starting material, reacted for 3 h.

**^1^H-NMR (400 MHz, CDCl_3_, 25 °C) δ/ppm:** 8.03 (d, 1H, J = 8.0 Hz), 7.91 (d, 1H, J = 7.4 Hz), 7.74 (t, 1H, J = 7.6 Hz), 7.70–7.62 (m, 3H,), 7.58–7.42 (m, 9H), 7.21 (t, 1H, J = 7.6 Hz), 7.07–6.91(m, 5H), 6.68(d, 1H, J = 7.6 Hz), 5.84 (s, 1H), 3.01 (d, 1H, J = 11.2 Hz), 2.87 (dd, 1H, J^1^ = 16.8 Hz, J^2^ = 11.2 Hz), 2.31 (dd, 1H, J^1^ = 16.8 Hz, J^2^ = 2.0 Hz).

**^13^C-NMR (100 MHz, CDCl_3_, 25 °C) δ/ppm:** 201.2, 197.1, 194.3, 142.8, 141.2, 138.6, 137.6, 137.3, 136.5, 135.9, 134.4, 133.3, 133.2, 132.0, 129.4, 129.1, 128.7, 128.1, 127.9, 127.8, 127.6, 127.1, 126.8, 125.2, 123.06, 123.00, 66.2, 66.0, 37.3, 36.4.

**HRMS** (ESI) for C_37_H_26_BrNO_5_S, [M + H]^+^ (676.0788), found 676.0783.

**IR** (KBr) ῡ (cm^−1^): 3064, 1741, 1706, 1585, 1357, 1250, 1170, 1071, 757, 582.

**3k** (123.4 mg, yield 96%, mp 269–270 °C): **1d** (81.7 mg, 0.2 mmol) and **2a** (46.9 mg, 0.2 mmol) as starting material, reacted for 3 h.

**^1^H-NMR (400 MHz, CDCl_3_, 25 °C) δ/ppm:** 8.21 (d, 2H, J = 8.8 Hz), 8.04 (d, 1H, J = 7.8 Hz), 7.93 (d, 1H, J = 7.6 Hz), 7.83–7.74 (m, 3H), 7.74–7.64 (m, 3H), 7.61–7.43 (m, 5H), 7.22 (t, 1H, J = 7.4 Hz), 7.03–6.95 (m, 5H), 6.60 (d, 1H, J = 7.6 Hz), 5.84 (s, 1H), 3.08–2.94 (m, 2H), 2.39 (d, 1H, 15.6 Hz).

**^13^C-NMR (100 MHz, CDCl_3_, 25 °C) δ/ppm:** 201.2, 197.0, 193.8, 150.5, 142.8, 141.2, 140.1, 138.5, 137.8, 137.4, 136.7, 136.0, 133.2, 133.0, 129.2, 129.0, 128.2, 128.10, 127.98, 127.7, 127.2, 126.82, 126.79, 124.9, 123.9, 123.2, 123.1, 66.3, 66.0, 37.1, 36.9.

**HRMS** (ESI) for C_37_H_26_N_2_O_7_S, [M + H]^+^ (643.1533), found 643.1588.

**IR** (KBr) ῡ (cm^−1^): 3429, 3075, 1706, 1594, 1524, 1346, 1251, 1172, 1089, 735, 558.

**3l** (111.8 mg, yield 89%, mp 219–220 °C): **1e** (78.7 mg, 0.2 mmol) and **2a** (46.9 mg, 0.2 mmol) as starting material, reacted for 12 h.

**^1^H-NMR (400 MHz, CDCl_3_, 25 °C) δ/ppm:** 8.02 (d, 1H, J = 8.2 Hz), 7.91 (d, 1H, J = 7.4 Hz), 7.75 (t, 1H, J = 7.4 Hz), 7.71 (d, 2H, J = 7.8 Hz), 7.66 (t, 1H, J = 7.4 Hz), 7.61 (d, 2H, J = 8.8 Hz), 7.58–7.43 (m, 5H), 7.20 (t, 1H, J = 7.6 Hz), 7.06–6.94 (m, 5H), 6.83 (d, 2H, J = 8.8 Hz), 6.70 (d, 1H, J = 7.8 Hz), 5.86 (s, 1H), 3.83 (s, 3H), 3.05 (d, 1H, J = 11.2 Hz), 2.88 (dd, 1H, J^1^ = 16.6 Hz, J^2^ = 11.2 Hz), 2.29 (dd, 1H, J^1^ = 16.6 Hz, J^2^ = 2.0 Hz).

**^13^C-NMR (100 MHz, CDCl_3_, 25 °C) δ/ppm:** 201.3, 197.3, 193.6, 163.7, 142.7141.3, 138.8, 137.6, 137.3, 136.4, 135.8, 133.7, 133.2, 130.2, 129.2, 128.9, 128.1, 127.8, 127.7, 127.5, 127.1, 126.8, 125.4, 123.0, 123.1, 113.8, 66.1, 66.0, 55.5, 37.5, 36.0.

**HRMS** (ESI) for C_38_H_29_NO_6_S, [M + H]^+^ (628.1788), found 628.1800.

**IR** (KBr) ῡ (cm^−1^): 3430, 3066, 1706, 1685, 1599, 1354, 1251, 1169, 735, 582.

**3m** (110.2 mg, yield 85%, mp 250–251 °C): **1g** (82.7 mg, 0.2 mmol) and **2a** (46.9 mg, 0.2 mmol) as starting material, reacted for 6 h.

**^1^H-NMR (400 MHz, CDCl_3_, 25 °C) δ/ppm:** 8.08 (s, 1H), 8.04 (d, 1H, J = 7.6 Hz), 7.93 (d, 1H, J = 7.6 Hz), 7.89 (d, 1H, J = 7.8 Hz), 7.81 (dd, 2H, J^1^ = 12.0 Hz, J^2^ = 8.2 Hz), 7.76–7.67 (m, 2H), 7.64–7.50 (m, 5H), 7.50–7.43 (m, 2H), 7.28–7.19 (m, 3H), 7.14 (t, 1H, J = 7.4 Hz), 7.06 (d, 1H, J = 6.6 Hz), 7.02–6.89 (m, 4H), 5.89 (s, 1H), 3.10 (d, 1H, J = 11.2 Hz), 3.01 (dd, 1H, J^1^ = 15.6 Hz, J^2^ = 11.2 Hz), 2.54 (dd, 1H, J^1^ = 15.6 Hz, J^2^ = 2.0 Hz).

**^13^C-NMR (100 MHz, CDCl_3_, 25 °C) δ/ppm:** 201.2, 197.0, 195.5, 142.7, 141.2, 138.7, 137.4, 137.2, 136.5, 135.9, 135.5, 133.5, 132.9, 132.8, 132.2, 129.7, 129.5, 128.9, 128.7, 128.5, 128.0, 127.8, 127.6, 127.5, 126.81, 126.77, 126.67, 125.4, 123.5, 122.9, 66.1, 66.0, 37.8, 36.6.

**HRMS** (ESI) for C_41_H_29_NO_5_S, [M + H]^+^ (648.1839), found 648.1846.

**IR** (KBr) ῡ (cm^−1^): 3430, 3063, 1741, 1706, 1682, 1593, 1356, 1251, 1170, 1091, 755, 733, 582.

### 4.4. Cell Culture

HaCaT cells which were kindly provided by Prof. Hung-Rong Yen (Integration of Traditional Chinese-Western Medicine Research Institute, China Medical University Hospital, Taiwan) were maintained in a high-glucose Dulbecco’s modified Eagle medium (DMEM), which was supplemented with 10% fetal bovine serum (FBS) and 1% penicillin and streptomycin. Cells were cultured at 37 °C in a humidified environment of 5% CO_2_. Then, 0.25% Trypsin-EDTA was used to vaccinate when the amount of cultured cells reached 1 × 10^6^ [32].

### 4.5. Cell Viability Assay

Cytotoxicity was analyzed using the MTT assay, with slight modifications. Briefly, HaCaT cells were seeded in 96-well plates to allow for overnight adhesion. The following day, cells were treated with various concentrations (0, 6.25, 12.5, 25, 50, and 100 µM) of each compound for 24 h. After treatment, MTT (100 µL/mL) was added to each well and incubated for 1 h at 37 °C under 5% CO_2_, and then DMSO (100 µL) was added to dissolve the formazan in the cells. The absorbance at 575 nm was measured using a microplate spectrophotometer. The results are expressed as a percentage of the corresponding controls [33].

### 4.6. Scratch Assay

HaCaT cells using a culture-insert (μ-Dish35mm, high, BLOSSOM BIOTECHNOLOGIES INC, Taipei, Taiwan) seeded in the precoated 24-well plates each compartment of the insert filled with 100 µL of cell suspension (about 2 × 10^5^ cells/chamber) for 24 h at 37 °C and 5% CO_2_. The cell debris was removed by washing with PBS when the cells grow to 80% confluence. Serum medium (control) and medium containing each tested compound (6.25, 12.5, 25, 50, and 100 µM) were added to corresponding wells, and the cells were cultured with serum supplemented DMEM for 36 h at 37 °C with 5% CO_2_ [34,35]. All scratch assays were conducted in triplicate. The areas of cell migration were analyzed using ImageJ software.

The wound closures were evaluated using the following formula:Wound closure (%) = 100 − [(sample area/control area) × 100].

### 4.7. Splints

The outer diameter, inner diameter, and thickness of round silicone splints were 26, 16, and 500–600 mm, respectively. All procedures and instruments were conducted and prepared, respectively, under aseptic conditions, which were maintained using autoclaves, ethylene oxide gas, 70% ethanol, and povidone-iodine. Mice were anesthetized with pentobarbital. An electric razor was used to remove the back hair, and a circular mark (1.0 cm diameter) was placed on the center of the lumbar area; this section of skin was totally excised using scissors. A splint was inserted beneath the skin near the wound defect and attached to the fascia with 6-stitch ligations. The splint was then fixed to the skin with surgical silk thread (6 stitches at regular intervals) [36,37].

### 4.8. Preparation of Sample-Containing PEG-Based Ointment

PEG 400 Da (100 mg) and PEG 4 kDa (20 mg) were mixed at a ratio of 5:1. The mixture was then heated to above 85 °C until it became a clear solution. The prepared compound (1 mg) was added to the solution before the solution was allowed to cool to room temperature, forming a gel [38].

### 4.9. Animal Experiments

Compounds were tested in 8-week-old wild-type male C57BL/6 mice (BioLASCO Taiwan Co., Ltd., Taipei, Taiwan). The mice were artificially wounded using splints. The control gel or the compound-containing gel was applied daily and photographed to measure the wound area for 13 days. The mice were sacrificed by CO_2_ exposure, and the tissues at the location of the silicone ring were removed for subsequent analysis.

### 4.10. Statistical Analysis

Analyses were performed in triplicate, and the results were expressed as mean ± SD. Analysis of variance (ANOVA) was conducted, followed by Dunnett’s post hoc test, to determine significant (*p* < 0.05) differences. Statistical analyses were performed using GraphPad Prism v8.0 (GraphPad Software, San Diego, CA, USA).

## 5. Conclusions

In this article, we developed a one-pot reaction method that requires only a small amount of catalyst to obtain nonselective isomers with high selectivity. In addition, the experimental procedure is simple and only requires extraction and recrystallization. The reaction was performed at room temperature and had a good yield. This reaction will be studied in our further research by using different catalysts. We hope that the synthesis strategy described in this article can be widely applied to various starting materials with diverse substituents. We also hope that the synthesis of *spiro*-tetrahydroqunioline derivatives could provide a new method for identifying chemicals that can be applied to wound-healing research. Further structural modifications and biological evaluations are ongoing, and the results will be reported in due course.

## Data Availability

Not applicable.

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
