# Peer review of "Synthesis of Novel Spiro-Tetrahydroquinoline Derivatives and Evaluation of Their Pharmacological Effects on Wound Healing"

_ijms, 2021, doi:10.3390/ijms22126251_

Round 1
Reviewer 1 Report
Title
There is already a typo in the title : tetrahydroquinoline (instead of qunioline)
Introduction
The rationale for the synthesis of spiro-tetrahydroquinolines and to study them as wound healing agents is not very strong, unless that both scaffolds are endowed with a plethora of biological activities. Is there any particular reason to have a sulfonamide moiety (either biologically or synthetically?)
Synthesis
Table 1 : only one diastereomer of 3a is shown. Is this the major isomer ? better to show that a mixture was formed and indicate the ratio of both isomers
Table 1 : the Xray of compound 3a is drawn on a weird place ; make a separate figure from this Xray structure
Is each diastereomer still a mixture of two enantiomers ? please comments
Has there been any efforts to separate both diastereomers ? Were they tested as biologically as mixtures (not clear from the experimental section) ?
The NMR data in experimental section : are these for the mixtures or for the major isomer ?
Xray experiment : was this done on a diastereomeric mixture ? Only one isomer is shown? What about the enantiomers in this Xray analysis.
For Xray analysis, the authors introduced reference 26 : however, reference 26 deals with peptides and has nothing to do with Xray
Line 78 : introduce a space between solubility and of
Line 78 : introduce a space between choose and to
Line 122 : is present rather than an aldehyde or ketone (instead of … than that …)
Biological evaluation
Table 3 and Figure 1 are the same data, presented in a different way. Move figure 1 to the supporting info?
Table 3 : the legend says inhibition of cell growth, whereas on the top of the table cell viability is mentioned. These are just the opposite. Also, on the top of the table, it says IC50, whereas no IC50 values are calculated. The correct wording is : % cell viability
Table 3/Fig 1 : why did the authors only run a cytotox assay for 24h? this is really short ; usually this type of assays are run 2-5 days.
Authors should include a positive control in fig 1 / table 3
The control as mentioned in Fig1/Table 3 : is this a negative control ? DMSO, buffer, PBS, …
Legend of Table 3 : determined (instead of detemined)
Legend of Table 3: keratinocyte (instead of kerathocyte)
Legend of Figure 1 : Delete : The cells were Table 24h
The authors claim 3i not to be cytotox, but, it seems one of the most toxic compounds ??
The quality/resolution of Fig 2 needs to be improved
In the text, authors should explain/elaborate on the scratch/migration assay, so that the reader can understand the basics of this assay
Figure 3 and 4 : can the authors include positive controls ?
Discussion
Discussion of the reaction mechanism should be moved to the chemistry section. Although the authors describe the two Michael addition, the rationale for preferential formation of one of the diastereomers is lacking.
The authors argue that compound 3i promotes would healing by promoting cell proliferation, but according to the MTT assay, it seems it is an inhibitor of cell proliferation?
Author Response
All line numbers mentioned below correspond to the document “Manuscript”.
Reviewer 1:
Title: There is already a typo in the title: tetrahydroquinoline (instead of qunioline)
Answer: We have corrected the words in title.
Introduction: The rationale for the synthesis of spiro-tetrahydroquinolines and to study them as wound healing agents is not very strong, unless that both scaffolds are endowed with a plethora of biological activities. Is there any particular reason to have a sulfonamide moiety (either biologically or synthetically?)
Answer: We have updated the pharmacological literature of quinoline in the introduction section. Quinolines are intensively studied as wound healing target compounds [ref. 9-14], however there is few report about tetrahydroquinoline on wound healing which is the reduced form of quinoline, even though their structures are so similar to each other. Besides, the synthesis of tetrahydroquinolines is one of the research topics in our ongoing projects for a long time, as a result we are curious that whether it will deliver similar effects on wound healing or not.
There are two reasons to have a sulfonamide moiety in the target compounds. Firstly, sulfonamide can be used as a protecting group for amino moiety. The presence of the sulfonamide group is essential for the reaction selectivity of the amino group. Without the sulfonamide group, a free amino group (-NH2) will react with the ketone moiety of 1a, and then the aza-Michael addition of the nitrogen to the enone moiety of 2a will be retarded. Secondly, in the period of the intramolecular Michael addition when cyclizing, the steric effect of the sulfonamide and enone moiety of 2a are important for leading to the diasterioselectivity (please see the proposed transition states below for illustrating the diastereoseletivity).
Synthesis:
Table 1: only one diastereomer of 3a is shown. Is this the major isomer? Better to show that a mixture was formed and indicate the ratio of both isomers.
Answer: In our all reactions, only one diastereomer was formed. Reasons for the diastereoselectivity were illustrated in the following figure which was based on the proposed transition states.
As shown in the figure above, the transition states have two isomers (syn- and anti-isomers) that can interchange with each other. In the piperidine ring, if the green hydrogen atom heads upward (syn-isomer), the steric effect of the sulfonamide and enone moiety was less impact, and compound 3 was easier formed. On the contrary, green enone moiety heads upward in the anti-isomer which shows more steric effect and makes it more difficult to form 3’.
Table 1: the X-ray of compound 3a is drawn on a weird place; make a separate figure from this X-ray structure.
Answer: We have made a separate figure for the X-ray structure of compound 3a in Figure 1.
Is each diastereomer still a mixture of two enantiomers? Please comments.
Answer: Yes, each product is still a mixture of two enantiomers, because we don’t employ any chiral reagent, catalyst, or auxiliary to conduct our reactions. That is why we only claim that our reaction has high diastereoselectivity instead of enantioselectivity.
Has there been any effort to separate both diastereomers? Were they tested as biologically as mixtures (not clear from the experimental section)?
Answer: Because only one diastereomeric product is found in each reaction, every biological test is based on the single diastereomer (3a to 3m).
The NMR data in experimental section: are these for the mixtures or for the major isomer?
Answer: Each NMR spectrum of compound 3 in supplement data is referring to the single diastereomer (syn-isomer).
X-ray experiment: was this done on a diastereomeric mixture? Only one isomer is shown? What about the enantiomers in this X-ray analysis.
Answer: The X-ray was done on single diastereomer.
For X-ray analysis, the authors introduced reference 26: however, reference 26 deals with peptides and has nothing to do with X-ray.
Answer: We have revised this section to reference 31.
Line 78: introduce a space between solubility and of
Answer: We have revised this section.
Line 78: introduce a space between choose and to
Answer: We have revised this section.
Line 122: is present rather than an aldehyde or ketone (instead of … than that …)
Answer: We have revised this section.
Biological evaluation
- Table 3 and Figure 1 are the same data, presented in a different way. Move figure 1 to the supporting info?
Answer: We have revised this section and moved Figure 1 to the supporting information.
- Table 3: the legend says inhibition of cell growth, whereas on the top of the table cell viability is mentioned. These are just the opposite. Also, on the top of the table, it says IC50, whereas no IC50 values are calculated. The correct wording is : % cell viability
Answer: Thank you for your valuable suggestion, and we have revised this section.
- Table 3/Fig 1: why did the authors only run a cytotox assay for 24h? This is really short ; usually this type of assays are run 2-5 days.
Answer: From our assays, these compounds showed both a clear range of inhibiting effects on HaCaT cells and easily recognized activities on cell migration in 24h, and the results could be obviously and satisfied observed. Furthermore, similar protocols were also used in some recent literatures listed below, and finally we considered that 24h-test was a suitable method for our experiment.
- Meng, Y.Q.; Tong, H.; Li, X.X.; Kuai, Z.Y.; Li, Q.W.; Xu, C.D. Synthesis and anti-tumor activity of derivatives of ring A of asiatic acid. J Asian Nat Prod Res. 2020, 22(7), 689-700.
- Popiołeka, L.; Patrejkoa, P.; Grzywaczb, G.M.; Biernasiukc, A.; Rycerzd, A.B.; Chomickab, D.N.; Chmielb, I.P.; Gumieniczekd, A.; Dudkab, J.; Wujeca, M. Synthesis and in vitro bioactivity study of new hydrazide-hydrazones of 5-bromo-2-iodobenzoic acid. Biomed Pharmacother. 2020, 130:110526.
- Imtiaz, Y.; Tuga, B.; Smith, C.; Rabideau, A.; Nguyen, L.; Liu, Y.; Hrapovic, S.; Ckless, K.; Sunasee, R. Synthesis and Cytotoxicity Studies of Wood-Based Cationic Cellulose Nanocrystals as Potential Immunomodulators. Nanomaterials (Basel). 2020, 10(8), 1603.
- Authors should include a positive control in fig 1 / table 3
Answer: In this manuscript, we focused on the wound healing effects of our 13 synthetic compounds. Because they were new compounds and the core structure of these compounds was not been studied on wound healing before, we cannot found a suitable positive control drug with comparably similar structures. Moreover, the wound healing effects of our compounds were apparently good to be observed in our assay despite we didn’t compare them with a clinical drug. Recently, some articles listed below also only used negative control to evaluate the effect of their compounds on wound healing, and therefore we considered that only using negative control was enough to evaluate our compounds. By the way, thank you for your suggestion and we will prepare more derivatives and build the SAR model to elute the more effective compounds with comparing a clinical drug as a positive control in the future.
- Kubica, K. P.; Taciak, P. P.; Czajkowska, A.; Ignasiak, A. S.; Wyrebiak, R.; Podsadni, P.; Bia£Y, I. M.; Malejczyk, J.; Mazurek, A. Synthesis and anticancer activity evaluation of some new derivatives of 2-(4-benzoyl-1-piperazinyl)-quinoline and 2-(4-cinnamoyl-1-piperazinyl)-quinoline. Acta Pol Pharm. 2018, 75(4), 891-901.
- Yang, J.D.; Moh, S.H.; Choi, Y.H.; Kim, K.W. b-Neoendorphin EnhancesWound Healing by Promoting Cell Migration in Keratinocyte. 2020, 25(20), 4640-4652.
- The control as mentioned in Fig1/Table 3: is this a negative control? DMSO, buffer, PBS, …
Answer: We used the medium for cell culture as our negative control.
- Legend of Table 3: determined (instead of detemined)
Answer: We have revised this section.
- Legend of Table 3: keratinocyte (instead of kerathocyte)
Answer: We have revised this section.
- Legend of Figure 1: Delete : The cells were Table 24h
Answer: We have revised this section.
- The authors claim 3i not to be cytotox, but, it seems one of the most toxic compounds??
Answer: 3i did have cytotoxicity. However, cell proliferation was not the only factor which influenced wound healing. So, we still wanted to realize the effect of compound 3i on the wound healing by carefully choosing suitable concentrations for operating the experiments.
- The quality/resolution of Fig 2 needs to be improved
Answer: We have revised the quality of Fig 2.
- In the text, authors should explain/elaborate on the scratch/migration assay, so that the reader can understand the basics of this assay
Answer: We have revised this section in the Materials and Methods.
- Figure 3 and 4: can the authors include positive controls?
Answer: As mentioned in question 4, we are not able to provide the results with a positive control.
Discussion
Discussion of the reaction mechanism should be moved to the chemistry section. Although the authors describe the two Michael additions, the rationale for preferential formation of one of the diastereomers is lacking.
Answer: We have revised this section. The position of the reaction mechanism had been revised. Besides, explain for preferential formation of only one diastereomer was included in the revised manuscript.
The authors argue that compound 3i promotes would healing by promoting cell proliferation, but according to the MTT assay, it seems it is an inhibitor of cell proliferation?
Answer: We have revised this section. Actually, wound healing was a multi-factor process, and proliferation and migration were two important factors in this process. As shown in the experimental results, the proliferation of cells was depressed due to the toxicity of compound 3i, but the migration effect of 3i was also apparently positive, and moreover 3i had positive effect on wound healing in the in vivo test. Finally, it was considered a candidate with positive potential effect on wound healing.

Reviewer 2 Report
The authors reported the synthesis of new Spiro-Tetrahydroquinoline Derivatives, applied to wound healing.
The synthetic method is already known, but well performed in the absence of chiral catalysts. The pharmacological properties seem well performed and described.
Some issues should be addressed prior to publication.
Please check comments below:
Introduction – comments
In the title, Spiro-Tetrahydroqunioline is spelled incorrectly. Please correct.
Lines 41-42: it would be interesting for readers if the authors would be able to provide up-to-date references for the biological activities of tetrahydroquinolines
Reference 16 is not for the isolation of Fredericamycin A. Please provide the reference reporting the isolation or biological activity of Fredericamycin A.
Line 48: Ref 18 is not illustrative of the attractiveness of spiro-1,3-indandiones, this ref is from 1996. Please include more recent reports.
Something is wrong with references 21, 22 and 23…They seemed in wrong places and inaccurately described. Please correct
The synthetic approach used by the authors is not new. Please check Org. Biomol. Chem. 2018, 16, 9390-9401. At least it should be cited as the author used the same aza-Michael/Michael cascade reactions.
Results section – comments
Whenever the authors are referring to Table 1 different entries, this should always appear like this (Table 1, entry 1) and so on.
The yields should be written without any space between the number and the percentage symbol (ex. 70%)
Line 78 there is a mistake in “solubilityof”, should be “solubility of”, as well as “chooseto” in the same line
Line 78-79: The authors choose CH2Cl2 for temperature screening. In fact, the authors only tested two different temperatures, 0 and 30 °C. This can not be stated as a temperature screening.
Experimental section - comments
This section is not well organized. Sometimes the NMR introduction appears in bold, sometimes not. Please uniformize.
Please make a clear separation between the characterization of each compound and derivative. Otherwise is quite difficult to identify the compounds. Perhaps put the name of the compound in bold to separate the different compounds from their characterization.
Abbreviation section - comments
Why polyethylene glycol is in italic???
Sometimes the abbreviation for ethyl acetate is EtOAc and sometimes is EA?? It would be better always use EtOAc
Author Response
Reviewer 2:
In the title, Spiro-Tetrahydroqunioline is spelled incorrectly. Please correct.
Answer: We have revised the title.
Lines 41-42: it would be interesting for readers if the authors would be able to provide up-to-date references for the biological activities of tetrahydroquinolines.
Answer: We have updated the references about biological activities of tetrahydroquinolines [ref. 14-15].
Reference 16 is not for the isolation of Fredericamycin A. Please provide the reference reporting the isolation or biological activity of Fredericamycin A.
Answer: We have changed the reference to a new one about the isolation and biological activity of Fredericamycin A [ref. 19].
Line 48: Ref 18 is not illustrative of the attractiveness of spiro-1,3-indandiones, this ref is from 1996. Please include more recent reports.
Answer: We have updated the references about spiro-1,3-indandiones [Ref 22].
Something is wrong with references 21, 22 and 23…They seemed in wrong places and inaccurately described. Please correct.
Answer: We have revised the order of references [ref. 24-26].
The synthetic approach used by the authors is not new. Please check Org. Biomol. Chem. 2018, 16, 9390-9401. At least it should be cited as the author used the same aza-Michael/Michael cascade reactions.
Answer: Thank you for your valuable suggestion, and the suggested article (Org. Biomol. Chem. 2018, 16, 9390-9401) had been quoted in the revised manuscript. Besides, we also added several articles to update this paragraph [ref.30].
Whenever the authors are referring to Table 1 different entries, this should always appear like this (Table 1, entry 1) and so on.
Answer: We have revised this section.
The yields should be written without any space between the number and the percentage symbol (ex. 70%)
Answer: We have revised this expression.
Line 78 there is a mistake in “solubilityof”, should be “solubility of”, as well as “chooseto” in the same line.
Answer: We have revised the two sections.
Line 78-79: The authors choose CH2Cl2 for temperature screening. In fact, the authors only tested two different temperatures, 0 and 30 °C. This cannot be stated as a temperature screening.
Answer: As your suggestion, we only chose two different temperatures for reactions, which cannot be stated as a temperature screening. So, we have revised the statement.

Round 2
Reviewer 1 Report
The authors addressed most of comments, with the exception of inclusion of a pos control in the biol assays
I assume that still the paper can be accepted